# Association between Suboptimal 25-Hydroxyvitamin D Status and Overweight/Obesity in Infants: A Prospective Cohort Study in China

**DOI:** 10.3390/nu14224897

**Published:** 2022-11-19

**Authors:** Chen Chen, Chunyan Zhou, Shijian Liu, Xianting Jiao, Xirui Wang, Yue Zhang, Xiaodan Yu

**Affiliations:** 1Department of Developmental and Behavioral Pediatrics, Shanghai Children’s Medical Center, School of Medicine, Shanghai Jiao Tong University, Shanghai 200127, China; 2Translational Medicine Institute, Shanghai Children’s Medical Center, School of Medicine, Shanghai Jiao Tong University, Shanghai 200127, China; 3Department of Clinical Epidemiology and Biostatistics, Children Health Advocacy Institute, Shanghai Children’s Medical Center, School of Medicine, Shanghai Jiao Tong University, Shanghai 200127, China; 4School of Public Health, School of Medicine, Shanghai Jiao Tong University, Shanghai 200127, China; 5Department of Pediatric Cardiology, Xinhua Hospital, School of Medicine, Shanghai Jiao Tong University, Shanghai 200092, China; 6MOE-Shanghai Key Lab of Children’s Environmental Health, Xinhua Hospital, School of Medicine, Shanghai Jiao Tong University, Shanghai 200092, China

**Keywords:** 25-hydroxyvitamin D, overweight and obesity, infants, threshold effect, prospective cohort study

## Abstract

This study aimed to investigate whether 25-hydroxyvitamin D (25(OH)D) concentrations are correlated to overweight/obesity in infants and to explore a threshold of 25(OH)D. A total of 1205 six-month-old infants from two community hospitals in Shanghai were randomly recruited, and 925 of them were followed up at 12 months. Concentration of 25(OH)D, weight, and length were measured at two time points. Overweight/obesity was defined as a weight-for-length Z-score >97th percentile. The prevalence of overweight/obesity at 6 and 12 months was 6.88% and 5.26%, respectively. The occurrence of vitamin D (VitD) deficiency (<20 ng/mL) at 6 and 12 months was 6.56% and 2.05%, respectively. Concentration of 25(OH)D at the corresponding age was negatively associated with weight-for-length percentile (WLP) at both 6 (adjusted β: −0.14; 95% CI: −0.27, −0.02; *p* = 0.02) and 12 months (adjusted β: −0.22; 95% CI: −0.41, −0.02; *p* = 0.03), while the relationship between 25(OH)D at 6 months and WLP at 12 months was nonlinear, where 35 ng/mL was identified as an inflection point. Those with a concentration of 25(OH)D <35 ng/mL at 6 months had a higher risk of overweight/obesity (adjusted OR: 1.42; 95% CI: 1.06, 1.91; *p* = 0.02) compared to the group with a concentration of 25(OH)D ≥35 ng/mL. Moreover, a concentration of 25(OH)D <35 ng/mL at two time points significantly increased the risk of overweight/obesity at 12 months compared to the group with 25(OH)D concentration ≥35 ng/mL at two time points (adjusted OR: 2.91; 95% CI: 1.13, 7.46; *p* = 0.03). A suboptimal 25(OH)D concentration <35 ng/mL significantly increases the risk of overweight/obesity in infants.

## 1. Introduction

Overweight/obesity status has been associated with various chronic conditions, e.g., diabetes, nonalcoholic steatohepatitis, dyslipidemia, hypertension, and heart disease [1,2]. Since 1997, the World Health Organization (WHO) has declared overweight/obesity a major public health problem and a global epidemic [3]. However, the prevalence of overweight/obesity continues to increase among children and adults around the world, including in China [1,4,5]. Increasing evidence suggests that the origins of overweight/obesity in children and adults can be traced back to the first 1000 days from conception to 2 years of age [2,6]. Therefore, there has been an increasing focus on preventing overweight/obesity in the earlier stages of life [7].

Vitamin D (VitD) is a fat-soluble steroid hormone that has a crucial role in calcium and phosphorus homeostasis, bone mineralization, and bone mass acquisition throughout our entire life, especially during childhood [8,9]. According to previous studies, skin synthesis with solar activation accounts for 80%–90% of VitD in the human body, while the rest comes from supplements or food [10]. So far, many studies have reported that VitD deficiency is a risk factor for overweight/obesity [10], and the potential regulatory mechanisms of VitD deficiency in human adipose tissue include direct adiposity-related gene regulation, and indirect modulation of parathyroid hormone (PTH), calcium, and leptin [11,12]. Specifically, VitD binds to VDR exerting endocrine, autocrine, or paracrine actions on the adipose tissue, which will inhibit adipocyte differentiation and reduce fatty acid synthesis, reducing lipid accumulation in vacuoles and inducing apoptosis of maturing preadipocytes [9,13]. VitD deficiency can also increase PTH, and then PTH enhances the production of the active VitD metabolite 1,25(OH)_2_D, which promotes lipogenesis through increased calcium influx to adipocytes, leading to excess fat and increased body weight [11,14]. Additionally, in vitro studies in human adipose tissue samples have shown that vitamin D_3_ inhibits leptin secretion [15], and observational studies support an inverse association between vitamin D and leptin [16].

So far, most of the studies have focused on the impact of VitD deficiency on overweight/obesity in childhood or school-aged children [17,18,19], and a few have considered the influence of VitD deficiency in pregnancy and its effect on offspring overweight/obesity [20,21,22]. The first year of life is one key stage for preventing obesity; however, studies involving the impact of VitD status on the risk of overweight/obesity in infants remain unclear.

Based on the above, we hypothesize that VitD deficiency similarly increases the risk for overweight/obesity in infants and shows a persistent effect. We recruited six-month-old infants from two community hospitals in Shanghai, China, and followed them up at 12 months to study the association between 25(OH)D concentration and overweight/obesity and to investigate whether there is a suboptimal level of VitD that contributes to overweight/obesity.

## 2. Materials and Methods

### 2.1. Study Design and Population

A total of 1205 six-month-old infants from two community hospitals in Shanghai, China, were recruited between September 2018 and July 2021, and 925 of them (76.76%) were followed up at 12 months. Participants were excluded if they had acute malnutrition, chronic diseases, endocrinopathies, or congenital abnormalities [23]. Additionally, they were required to meet the following inclusion criteria: (1) their parents/guardians were continuous residents in Shanghai for at least 2 years; (2) parent/guardian provided written informed consent. This study was approved by the Ethics Committee of Shanghai Children’s Medical Center, School of Medicine, Shanghai Jiao Tong University (SCMCIRB-K2018061-1). The authors assert that all procedures complied with the ethical standards of the relevant national and institutional committees on human experimentation and were in accordance with the Helsinki Declaration [24]. This study followed the reporting guidelines defined by Strengthening the Reporting of Observational Studies in Epidemiology (STROBE) (https://www.strobe-statement.org/checklists/, accessed on 1 July 2018).

### 2.2. Data Collection by the Questionnaire

After enrollment, the experienced community pediatricians conducted a detailed face-to-face interview with each infant’s parent/guardian. Information on demographic characteristics, socioeconomic status, lifestyle habits, parental education, household income, number of children, passive smoking, feeding modality (exclusively breastfeeding was defined as infants receiving only breast milk before the age of 6 months, otherwise nonexclusively breastfeeding was confirmed), VitD supplements, outdoor time, and allergic disease were collected through questionnaires during the interview. Infant sex, birth date, birth weight, gestational age, and mode of delivery were abstracted from medical records.

### 2.3. Anthropometric Measurements

Skilled nurses performed anthropometric measurements in infants at 6 months and 12 months. Standardized supine position measurements of weight (the estimated weight of clothes and diapers was deducted) and length were obtained using qualified electronic measuring instruments (Seca 376 Baby scale, Seca, Hangzhou, China). Weight and length were accurate at 0.1 kg and 0.1 cm, respectively. The raw data of length and weight were converted to age- and sex-specific weight-for-length Z-score (WLZ) and weight-for-length percentile (WLP) using the World Health Organization (WHO) Anthro Software [25].

According to the WHO classification, overweight/obesity was defined as a WLZ >97th percentile and 99th percentile, respectively [26]. However, since the number of infants with obesity at 6 and 12 months was insufficient for reliable analysis (2/1205 and 3/925, respectively), infants with a WLZ >97th percentile were analyzed as a single category in this study.

### 2.4. 25(OH)D Measurement

A total of 20 µL of capillary blood was drawn from fingertips and collected in a sterile Eppendorf tube containing anticoagulant ethylene diamine tetraacetic acid [27]. After centrifugation, serum samples were immediately placed in −20 °C freezers away from light. The serum 25(OH)D measurement was performed as previously described [27]. Briefly, a high-performance liquid chromatography–tandem mass spectrometry (HPLC–MS/MS) (AB Sciex4500MD, ABSciex Company, Framingha, Massachusetts, USA) was applied for the measurement of 25(OH)D_3_. All blood specimens were measured in duplicates. Finally, the serum 25(OH)D_3_ concentration value was corrected with a transformation formula previously developed by our research team [27].

Based on the recommendations of the Endocrine Society, 25(OH)D levels <20 ng/mL indicated deficiency, while 20–29 ng/mL and ≥30 ng/mL indicated insufficiency and sufficiency, respectively [28].

### 2.5. Statistical Analysis

Quantitative variables were tested for normal distribution with the Kolmogorov-Smirnov test. Normal continuous variables were expressed as mean ± standard deviation (SD), and categorical variables were expressed as numbers and percentages. For group comparisons, independent *t*-tests or one-way ANOVA (continuous data) and the chi-square test (categorical data) were applied. Univariate analysis was performed to examine the association between the factors and WLP obtained at two time points. Smooth curve fitting was employed to display the relationship between 25(OH)D concentrations and WLP. According to the results of the smoothing plot, the segmented regression model of threshold effect was further applied to determine if there was a piecewise linear relationship. Multivariable linear regression was used to investigate the associations between the 25(OH)D concentration and WLP. Finally, a multiple logistic regression model was applied to estimate the independent impact of 25(OH)D concentration on the risk of overweight/obesity with adjustment for infant sex, birth weight, gestational age, maternal education, feeding modality, and allergic disease, which were selected a priori or identified in univariate analyses. All *p*-values were a two-tailed test, and *p* < 0.05 indicated a statistically significant difference. All analyses were conducted in Empower (R) (www.empowerstats.com accessed on 31 December 2020, X&Y solutions, Inc., Boston, MA, USA) and R 3.5.1 (http://www.R-project.org, accessed on 31 December 2020).

## 3. Results

### 3.1. Demographic Characteristics

The characteristics of WLP distribution and 25(OH)D status in infants at 6 and 12 months were analyzed by dividing WLP into four subgroups (<3; ≥3, <50; ≥50, <97; ≥97) and 25(OH)D status into three categories (<20; ≥20, <30; ≥30). As shown in Table 1, there were 6.88% and 5.26% of overweight/obese infants at 6 and 12 months, respectively. The median 25(OH)D concentration was 40.6 ng/mL and 42.3 ng/mL at the two time points, respectively. The prevalence of 25(OH)D deficiency was 6.56% at 6 months and 2.05% at 12 months, while the prevalence of 25(OH)D insufficiency was 11.37% at 6 months and 8.97% at 12 months. Participants’ characteristics are displayed in Table 2. We considered many factors possibly related to overweight/obesity in the present study. The univariate analysis revealed that 25(OH)D concentrations at 6 months were associated with WLP at 6 and 12 months. Infants at 6 months in the first quartile of 25(OH)D concentration had significantly higher WLP at both monitored intervals (*p* = 0.04 at two time points). Similarly, the 25(OH)D deficiency and insufficiency groups were associated with a higher WLP compared to the sufficient group (*p* = 0.02 at 6 months; *p* = 0.01 at 12 months). All other factors were not associated with WLP at the two time points (all *p* > 0.05).

### 3.2. Association between 25(OH)D Concentrations and Weight-for-Length Percentile

Based on the results of univariate analysis, we further determined the correlation between 25(OH)D concentrations and WLP using generalized additive models. As shown in Figure 1A,C, 25(OH)D levels at two time points were all negatively correlated with WLP at the corresponding age point. At 6 months, the inverse relationship revealed a 0.14 reduction of WLP for each 1 ng/mL increment in 25(OH)D concentration (adjusted β: −0.14; 95% CI: −0.27, −0.02; *p* = 0.02) (Table 3). In addition, a negative correlation was discovered between 25(OH)D concentration at 12 months and WLP at 12 months (adjusted β: −0.22; 95% CI: −0.41, −0.02; *p* = 0.03).

The association between 25(OH)D level at 6 months and WLP at 12 months was nonlinear and displayed a threshold effect (Figure 1B). When 25(OH)D concentration was lower than the inflection point of 35 ng/mL, 25(OH)D concentration was inversely correlated with WLP (adjusted β: −0.39; 95% CI: −0.70, −0.07; *p* = 0.02). In contrast, no significant relationship was found when 25(OH)D concentration was higher than the inflection point (adjusted β: 0.10; 95% CI −0.11, 0.31; *p* = 0.35) (Table 4).

### 3.3. Association between 25(OH)D Levels and Risk of Overweight/Obesity

Infants were divided into two subgroups on the basis of the threshold value (35 ng/mL), and a multivariate logistic regression model was used to explore the impact of 25(OH)D levels on the risk of overweight/obesity; the results are shown in Table 5. Compared with 25(OH)D concentration ≥35 ng/mL at 6 months, there was a significant association between 25(OH)D concentration <35 ng/mL and the risk of overweight/obesity at 6 months (adjusted odds ratio (OR): 1.42; 95% CI 1.06, 1.91; *p* = 0.02). There was no significant relationship between 25(OH)D below 35 ng/mL at 6 months and the risk of overweight/obesity at 12 months (adjusted OR: 1.05; 95% CI 0.74, 1.48; *p* = 0.80). Likewise, 25(OH)D <35 ng/mL at 12 months had no significant effect on the risk of overweight/obesity at 12 months (adjusted OR: 1.44; 95% CI 0.47, 4.39; *p* = 0.52).

We further studied the joint effect of 25(OH)D levels at 6 months (indicated as T1) and 12 months (indicated as T2) on the risk of overweight/obesity at 12 months (Table 6). Taking both T1 and T2 >35 ng/mL as a reference, when both T1 and T2 were <35 ng/mL, the risk of overweight/obesity at 12 months was significantly increased (adjusted OR: 2.91; 95% CI 1.13, 7.46; *p* = 0.03); while with either T1 or T2 <35 ng/mL, no increased risk of overweight/obesity at 12 months was observed (adjusted OR: 0.44; 95% CI 0.12, 1.56; *p* = 0.20).

## 4. Discussion

To the best of our knowledge, this prospective cohort study with a relatively large sample size is the first to investigate the relationship between 25(OH)D levels and WLP in infants at 6 and 12 months, and it confirmed that low VitD concentrations increase the risk of overweight/obesity in infants. Moreover, a suboptimal level of 25(OH)D during the first year of life may be relevant for infant overweight/obesity.

In our study, the prevalence of overweight/obesity in infants was 6.88% at 6 months and 5.26% at 12 months. A meta-analysis study in China reported that the overweight and obesity rates in infancy from 1991 to 2015 were 11.7% and 7.0%, respectively [5]. In a retrospective study from Anhui province, China, the overweight and obesity rates at 6 and 12 months were 7.1% and 6.1%, respectively [29]. In the Czech, 3.0% and 2.5% of infants were reported to be overweight and obese at 6 months and 12 months between 2008 and 2011 [30], while in a Canadian study, 6.5% of infants at one year were overweight in 2018 [31]. The overweight/obesity prevalence in infants has decreased over recent years; however, the current rates are still relatively high. So far, many risk factors contributing to overweight/obesity in early life have been confirmed and included in strategies to prevent overweight/obesity in later life, e.g., parental education, income, mode of delivery, and birth weight [32], but this has not included VitD status. The impact of VitD status on overweight/obesity in early life needs to be further investigated, as this may optimize the strategies to decrease overweight/obesity rates in both early and later life.

Levels of 25(OH)D and VitD deficiency/insufficiency prevalence tend to vary in infants from different cities or countries. In our study, the median concentrations at 6 and 12 months were 40.6 ng/mL and 42.3 ng/mL, while the prevalence of VitD insufficiency and deficiency was 17.93% and 11.02%, respectively. A previous study conducted in Huzhou city of China reported that the mean 25(OH)D levels at 4–6 months and 7–12 months were 43 ng/mL and 45 ng/mL, and the corresponding prevalence of VitD insufficiency or deficiency in the two age groups was 10.92% and 8.94%, respectively [33]. In another study from Hong Kong, China, the mean 25(OH)D level at 2–6 months was 23.8 ng/mL, and the frequency of insufficient and deficient VitD status even reached 37.1% [34]. In a study conducted in Turkey, the mean 25(OH)D concentrations in infants <6 months and 12 months were 28.18 ng/mL and 28.45 ng/mL, respectively; however, 25.5% of infants had an insufficient or deficient VitD status [35]. Moreover, 25(OH)D concentration <15 ng/mL was considered a deficiency in this Turkish study.

The above studies provide the latest data on VitD status in infancy. Although risk factors for VitD deficiency/insufficiency in infants, such as limited sun exposure, age, and maternal vitamin D status, have been well studied [34,36], VitD deficiency/insufficiency in infants is still a common occurrence. Moreover, as one of the important nutrients, VitD deficiency in early life has been implicated in many diseases, such as allergic diseases, growth retardation, and some nervous system disorders [37,38,39]. Hence, VitD status in infancy needs to be given more attention.

VitD status significantly affects infant physical growth, and low VitD status can remarkably increase the risk of overweight/obesity. The weight-for-length Z-score is a predominant standard for assessing overweight/obesity in children younger than 2 years old [40]. In our study, VitD status was not only significantly related to WLP but also had a strong negative linear relationship with WLP at 6 and 12 months in infants. Moreover, six-month-old infants with 25(OH)D levels <35 ng/mL had a 1.42-fold increased risk of overweight/obesity, while the risk in infants with 25(OH)D levels <35 ng/mL at two time points was higher—2.91-fold. One cross-sectional study carried out in 2021 also found the same relationship, where the increased risk of infancy overweight/obesity was attributed to VitD deficiency; inverse linear relationships were observed between 25(OH)D level and body mass index as well as BMI z-score in one-year-old infants [35]. In this study, they found that infants with a deficient VitD status had a 2.74-fold increased risk for obesity [41]. Although there has been limited research on the impact of VitD on infant overweight/obesity, these findings are promising and warrant more investigation, such as extending the follow-up period or developing multicenter trials across cities.

We found 25(OH)D had a threshold effect on overweight/obesity. Currently, the definition of VitD status is based on the Endocrine Society vitamin D guideline [42]. However, the recommended 25(OH)D levels were defined on the basis of studies focused on bone disease [42]. In our study, the suboptimal 25(OH)D level of <35 ng/mL for infants at 6 months significantly increased the risk for overweight/obesity at 12 months. There is considerable evidence that school-aged children and adolescents with a 25(OH)D level <20 ng/mL are susceptible to overweight/obesity [19,41,43]. Esmaili et al. pointed out that a higher level of 25(OH)D of 30 ng/mL was a risky inflection point [18]. Other studies set up suitable VitD levels for overweight/obesity or metabolism disorder in children on the basis of their study population, and inflection values varied, ranging from 10 ng/mL to 32 ng/mL [44,45]. In one review, the authors recommended maintaining serum 25(OH)D levels >30 ng/mL in pediatrics to prevent VitD deficiency, thus avoiding the risk of both skeletal and extraskeletal diseases [46]. Compared to earlier findings, the threshold of 35 ng/mL that we used was higher than that reported before, which may be explained by the high popularity of regular VitD supplementation (98.85%) in Shanghai, which has a high level of medical care. This discrepancy can also be attributed to the younger age of the study population. As it is difficult to agree on a recommended blood level of 25(OH)D, large, well-designed studies should be conducted to evaluate suboptimal 25(OH)D levels in relation to diverse ethnicity, age groups, and diseases.

Potential explanations for the existence of a 25(OH)D threshold with very different effects may be the different growth status of children and concentration-dependent characteristics of 25(OH)D. The effect of 25(OH)D might depend on BMI status. As was reported for 6–18-year-old children, when the BMI z-score was <0, BMI increased with an increasing 25(OH)D concentration, and BMI decreased with the 25(OH)D concentration if the BMI z-score was ≥0 [47,48]. Moreover, dual effects of VitD on modulating adipocytes were found in in vitro experiments, where low physiological 1α-25(OH)2D3 concentrations (10^−13^ and 10^−11^ mol/L) increased fat droplet accumulation, whereas high physiological (10^−9^ mol/L) and supraphysiological concentrations (≥10^−7^ mol/L) inhibited fat accumulation [48]. In addition, we speculate that the function of greater 25(OH)D concentration in infancy might be to maintain normal growth rather than having an important role in inhibiting adipose tissue formation or differentiation. Thus, it is plausible that we found a modest association with WLP at 12 months with 25(OH)D concentration over the suboptimal level at 6 months.

One strength of our study is that we focused on the effect of 25(OH)D in infancy on the risk of overweight/obesity. Moreover, we proposed a suboptimal level of 25(OH)D <35 ng/mL during the first year of life as an independent risk factor for infant overweight/obesity.

This study has some limitations. Firstly, our follow-up visit was only up to one year of age; we will continue tracking the long-term effects of VitD status. Secondly, although the rate of loss-to-follow-up was relatively high (23.24%), the rate was still acceptable due to the COVID-19 epidemic crisis. Thirdly, we did not consider some confounders, such as dietary intake and energy expenditure, which may bias our final results. Finally, our study was conducted in infants from Shanghai only, and our results need to be validated or improved in a large-population study involving more cities. Similarly, our findings and the suboptimal 25(OH)D levels varied in relation to genetics, age, and disease, so it is important to be cautious when extrapolating them to other populations or diseases.

## 5. Conclusions

With a relatively high prevalence of VitD deficiency and insufficiency, overweight/obesity was common in infants. VitD status in infants was significantly related to physical growth, and a suboptimal VitD status remarkably increased the risk of overweight/obesity. Levels of 25(OH)D had a threshold effect on overweight/obesity. The effects of VitD status in earlier stages of life on preventing obesity in later life should be further investigated.

## Figures and Tables

**Figure 1 nutrients-14-04897-f001:**
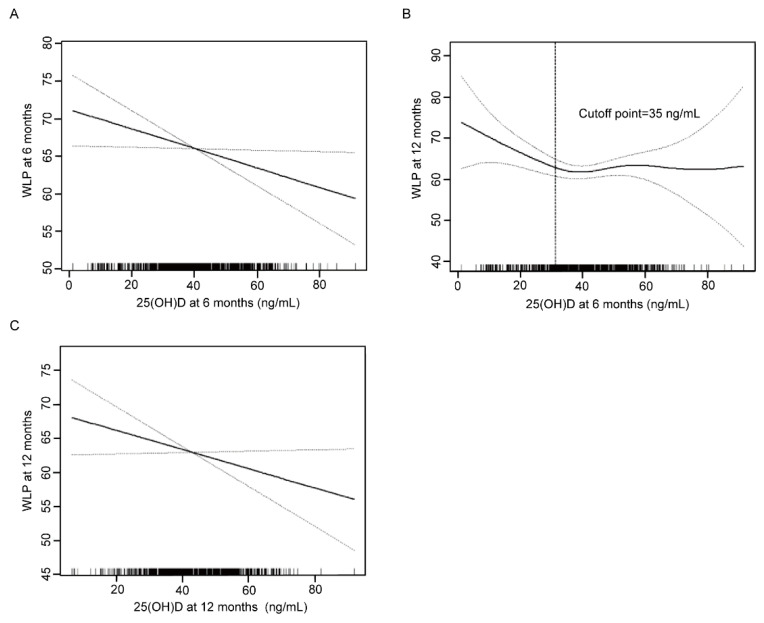
Association between 25(OH)D at 6 months and WLP at 6 months (**A**), 25(OH)D at 6 months and WLP at 12 months (**B**), 25(OH)D at 12 months and WLP at 12 months (**C**). Solid line represents the smooth curve fit between variables. Dotted line represents the 95% confidence interval from the fit. WLP, weight-for-length percentile.

**Table 1 nutrients-14-04897-t001:** Weight-for-length percentile distribution and 25(OH)D status in infants at 6 and 12 months.

Variables	6 Months (*n* = 1205)	12 Months (*n* = 925)
	*n* (%)	*n* (%)
Weight-for-length percentile		
<3	9 (0.77)	4 (0.45)
≥3, <50	282 (24.25)	261 (29.19)
≥50, <97	792 (68.10)	582 (65.10)
≥97	80 (6.88)	47 (5.26)
25(OH)D median (ng/mL)	40.6	42.3
25(OH)D categories (ng/mL)		
Deficiency (<20)	79 (6.56)	19 (2.05)
Insufficiency (≥20, <30)	137 (11.37)	83 (8.97)
Sufficiency (≥30)	989 (82.07)	823 (88.97)

**Table 2 nutrients-14-04897-t002:** Univariate analysis of factors associated with weight-for-length percentile (WLP) in infants at 6 and 12 months.

Variables	%	WLP at 6 Months		WLP at 12 Months	
		(*n* = 1205)		(*n* = 925)	
		Mean ± SD	*p*	Mean ± SD	*p*
25(OH)D at 6 months quartile (ng/mL)					
Q1 (<32.9)	24.81	69.29 ± 25.36	0.04 *	66.34 ± 25.55	0.04 *
Q2 (32.9–40.6)	24.98	64.52 ± 26.64		59.65 ± 26.00	
Q3 (40.6–48.6)	25.06	63.35 ± 26.67		61.89 ± 26.93	
Q4 (≥48.6)	25.15	65.96 ± 24.73		64.29 ± 25.07	
25(OH)D at 6 months categories (ng/mL)					
Deficiency and insufficiency (<30)	17.93	69.79 ± 25.58	0.02 *	67.98 ± 25.67	0.01 *
Sufficiency (≥30)	82.07	64.91 ± 25.93		62.03 ± 25.92	
25(OH)D at 12 months quartile (ng/mL)					
Q1 (<35.2)	24.54	65.15 ± 26.36	0.28	62.99 ± 25.24	0.06
Q2 (35.2–42.3)	25.3	68.14 ± 25.52		66.51 ± 24.96	
Q3 (42.3–51.1)	25.08	67.90 ± 25.23		62.48 ± 26.60	
Q4 (≥51.1)	25.08	64.25 ± 26.31		59.49 ± 26.40	
25(OH)D at 12 months categories (ng/mL)					
Deficiency and insufficiency (<30)	11.03	66.99 ± 25.61	0.80	64.54 ± 24.43	0.52
Sufficiency (≥30)	88.97	66.28 ± 25.92		62.71 ± 26.06	
Sex					
Boys	54.74	65.89 ± 26.25	0.95	62.53 ± 26.07	0.37
Girls	45.26	65.98 ± 25.38		64.08 ± 25.48	
Feeding modality					
Exclusively breastfeeding	53.02	66.80 ± 25.94	0.34	62.04 ± 26.02	0.18
Nonexclusively breastfeeding	46.98	65.27 ± 25.65		64.42 ± 25.64	
Dose of VitD supplementation (IU)					
≤400	55.85	66.58 ± 24.87	0.97	61.87 ± 25.78	0.11
>400	44.15	66.65 ± 25.60		64.97 ± 25.45	
Mode of delivery					
Natural delivery	58.42	65.43 ± 26.60	0.39	62.99 ± 26.01	0.55
Caesarean delivery	41.58	66.82 ± 25.03		64.08 ± 25.62	
Gestational age(weeks)					
<37	4.29	72.22 ± 21.82	0.11	61.08 ± 28.87	0.61
≥37	95.71	65.89 ± 25.96		63.44 ± 25.68	
Birthweight (kg)					
<2.5	2.57	65.34 ± 24.50	0.89	61.37 ± 23.52	0.84
≥2.5, <4	90.82	65.81 ± 25.92		62.91 ± 26.03	
≥4	6.61	67.29 ± 24.42		64.78 ± 24.06	
Allergy disease					
No	73.83	66.69 ± 25.75	0.13	63.48 ± 25.61	0.79
Yes	26.17	64.01 ± 25.93		62.96 ± 26.44	
Passive smoking					
No	89.58	65.90 ± 25.68	0.67	63.37 ± 25.75	0.63
Yes	10.42	67.00 ± 26.07		61.93 ± 25.87	
Outdoor time (h/d)					
<0.5	20.53	68.52 ± 24.36	0.35	62.65 ± 25.86	0.82
≥0.5, <1.0	36.68	65.61 ± 26.72		62.69 ± 25.75	
≥1.0, <2.0	32.48	65.64 ± 25.03		64.19 ± 25.32	
≥2.0	10.31	63.57 ± 26.65		61.70 ± 27.30	
Paternal education					
High school or below	9.39	64.12 ± 27.58	0.42	61.33 ± 25.41	0.52
College or above	90.61	66.26 ± 25.49		63.36 ± 25.83	
Maternal education					
High school or below	9.54	64.80 ± 27.33	0.62	63.11 ± 26.03	0.98
College or above	90.46	66.11 ± 25.61		63.18 ± 25.80	
Monthly income (CNY)					
<10,000	21.72	63.42 ± 26.45	0.21	60.37 ± 25.99	0.23
≥10,000, <20,000	45.62	66.36 ± 25.65		64.30 ± 25.78	
≥20,000	32.66	67.12 ± 25.38		63.30 ± 25.69	

* *p* < 0.05.

**Table 3 nutrients-14-04897-t003:** Associations of 25(OH)D with weight-for-length percentile in infants at 6 and 12 months.

Weight-for-Length Percentile	25(OH)D (ng/mL)	
	*β* (95% CI)	*p*
6 months (*n* = 1205)	25(OH)D at 6 months (ng/mL)	
Unadjusted	−0.15 (−0.26, −0.03)	0.01 *
Adjusted ^†^	−0.14 (−0.27, −0.02)	0.02 *
12 months (*n* = 925)	25(OH)D at 12 months (ng/mL)	
Unadjusted	−0.14 (−0.29, 0.01)	0.07
Adjusted ^†^	−0.22 (−0.41, −0.02)	0.03 *

* *p* < 0.05. ^†^ Adjusted for sex, birth weight, gestational age, maternal education, feeding modality, allergic disease.

**Table 4 nutrients-14-04897-t004:** Threshold effect analysis of 25(OH)D at 6 months and weight-for-length percentile in infants at 12 months using piecewise linear regression (*n* = 925).

Inflection of 25(OH)D (ng/mL)	Unadjusted		Adjusted ^†^	
	*β* (95% CI)	*p*	*β* (95% CI)	*p*
<35	−0.41 (−0.70, −0.11)	0.007 *	−0.39 (−0.70, −0.07)	0.02 *
≥35	0.10 (−0.10, 0.29)	0.33	0.10 (−0.11, 0.31)	0.35

* *p* < 0.05. ^†^ Adjusted for sex, birth weight, gestational age, maternal education, feeding modality, allergic disease.

**Table 5 nutrients-14-04897-t005:** Effects of 25(OH)D level on overweight/obesity in infants at 6 and 12 months.

Overweight/Obesity	25(OH)D Categories		
	Reference	OR (95% CI)	*p*
	25(OH)D range at 6 months (ng/mL)		
6 months	≥35 (*n* = 821)	<35 (*n* = 384)	
Unadjusted	1.00	1.32 (1.01, 1.72)	0.04 *
Adjusted ^†^	1.00	1.42 (1.06, 1.91)	0.02 *
12 months			
Unadjusted	1.00	1.16 (0.84, 1.61)	0.35
Adjusted ^†^	1.00	1.05 (0.74, 1.48)	0.80
	25(OH)D range at 12 months (ng/mL)		
12 months	≥35 (*n* = 701)	<35 (*n* = 224)	
Unadjusted	1.00	1.44 (0.64, 3.24)	0.38
Adjusted ^†^	1.00	1.44 (0.47, 4.39)	0.52

* *p* < 0.05. ^†^ Adjusted for sex, birth weight, gestational age, maternal education, feeding modality, allergic disease.

**Table 6 nutrients-14-04897-t006:** Joint effects of 25(OH)D level at two time points on the risk of overweight/obesity in infants at 12 months.

Model	T1 ^‡^ ≥ 35 and T2 ^‡^ ≥ 35	T1 ^‡^ ≥ 35 or T2 ^‡^ ≥ 35		T1 ^‡^ < 35 and T2 ^‡^ < 35	
	(*n* = 523)	(*n* = 256)		(*n* = 259)	
	Reference	OR (95% CI)	*p*	OR (95% CI)	*p*
Unadjusted	1.00	0.53 (0.17, 1.62)	0.26	2.31 (0.95, 5.60)	0.06
Adjusted ^†^	1.00	0.44 (0.12, 1.56)	0.20	2.91 (1.13, 7.46)	0.03 *

* *p* <0.05. ^†^ Adjusted for sex, birth weight, gestational age, maternal education, feeding modality, allergic disease. ^‡^ T1, 25(OH)D level at 6 months; T2, 25(OH)D level at 12 months.

## Data Availability

The data presented in this study are available on request from the corresponding author. The data are not publicly available because they contain information that could compromise the privacy of research participants.

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
