# Peer review of "Association between Suboptimal 25-Hydroxyvitamin D Status and Overweight/Obesity in Infants: A Prospective Cohort Study in China"

_nutrients, 2022, doi:10.3390/nu14224897_

Round 1
Reviewer 1 Report
Nutrients 198851
I would like to thank the editors for providing me the opportunity to review this manuscript.
The manuscript is well written, however, I would like to highlight the importance of dietary intake when talking about obesity.
I can see that the authors didn’t involve dietary intake as independent variable. Also, the main conclusion was the modulation effect of “vitamin D deficiency” on obesity, but without taking into consideration this confounding factor.
In line 100, “However, since the number of infants 100 with obesity at 6 and 12 months were insufficient for reliable analysis (2/1,205 and 3/925, 101 respectively), infants with WLZ > 97th percentile were analyzed as a single category in 102 this study.”: based on this sentence, I am wondering what is the effect of vitamin D deficiency on stunting and wasting?. The analysis of this population specifically, biased the results, the title and the conclusion. Why? Because the study was conducted among this population only!
Please revisit the methodology, and other part of the manuscript.
Author Response
Thank you for the time and effort that you have put into reviewing the previous version of the manuscript. Your suggestions have enabled us to improve our work. We have tried our best to revise our manuscript and provide the point-by-point responses according to your comments and suggestions.
Point 1: I can see that the authors didn’t involve dietary intake as independent variable. Also, the main conclusion was the modulation effect of “vitamin D deficiency” on obesity, but without taking into consideration this confounding factor.
Response 1: Thank you for your comment. Yes, many studies have revealed the relationship between dietary intake and overweight/obesity in children and adolescents, although the findings were mixed and inconsistent (PMID: 17341216, PMID: 17414504). So far, we haven’t found any study to investigate the association between dietary intake and overweight/obesity in infants, and our questionnaire only contains the feeding modality and does not involve the volume of breastmilk, food items consumed and energy intake, we cannot exclude the effect of dietary intake on the overweight/obesity in our participants. Generally, we agree that dietary intake as an independent variable should be considered, so this point as one of our study limitations has been added to the “Discussion” part (Line 301 and 302 on page 10) and will be explored in our future study.
Besides, we considered the dietary intake of vitamin D. We thought the different feeding modality may influence the different 25(OH)D concentrations in infants. We conducted a univariate analysis of feeding modality associated with 25(OH)D in infants, and we found the exclusively breastfeeding group had lower 25(OH)D than the non-exclusively breastfeeding group. So, we also adjusted the feeding modality when conducting multivariable analysis.
Point 2: In line 100, “However, since the number of infants 100 with obesity at 6 and 12 months were insufficient for reliable analysis (2/1,205 and 3/925, 101 respectively), infants with WLZ > 97th percentile were analyzed as a single category in 102 this study.”: based on this sentence, I am wondering what is the effect of vitamin D deficiency on stunting and wasting?. The analysis of this population specifically, biased the results, the title and the conclusion. Why? Because the study was conducted among this population only!
Response 2: Thank you for your comments. Since the purpose of the current study is to illustrate the impact of vitamin D deficiency on overweight/obesity in infants, we did not focus on the effect of vitamin D deficiency on stunting and wasting and will not discuss it in the manuscript either. However, we fully agree that vitamin D deficiency may be a potential risk factor for stunting and wasting in childhood (PMID:29162164, PMID:34150687). Then, we tried to analyze the association in our population. Unfortunately, the number of infants with wasting (WLP<P3: 9 (0.77%) and 4 (0.45%) at each time point) is insufficient for reliable analysis.
As you mentioned above, bias may arise as the analysis was performed in this population only. So far, this project has been finished, we are sorry that we cannot recruit infants from different cities to validate or strengthen our findings. Concerning the misinterpretation of our results, we added some contents in the “Discussion” part in the line 302 to 306 on page 10.
Reviewer 2 Report
In presented study authors try to show association between suboptimal 25-hydroxyvitamin D status and overweight/obesity in infants populations. Presented work is a prospective cohort study on population of China infants. Due the very important scope of presented work (Vitamin D status and its role in obesity in very young population i suggest some cahnges to achive Journal scopes.
Introduction
Authors write about the VitD role but not showing any asocieations of its concentraction and human body regulation (how it takeds place, what is it's role etc...but showed in mechnistic forme).
I suggest to showe what is the mains scop of this work and why it is writen - hypothesis and some sugestions????
Methodology
What was a tottal population of infants in presented population and how it was calculated that presented population is enough for the statistical significnte (1,205 six months old infants and 925 of them (76.76%) was falowed)?????
Why only one of vit D metabolite was messured, what with others???? In some cases it may not showe the hydroxylic activity in nonenzymatic skin hydroxylation and enzymatic reni hydroxylation. Morover I suggest that authors showe what are the associetation beetwen the serum concentraction and proposed tiny layer evaluation of VitD metabolite.
First, our follow-up visit was only up to one year of 287 age; we will continue tracking the long-term effects of VitD status. Second, although the 288 rate of loss-to-follow-up was relatively high (23.24%), the rate was still acceptable due to 289 the COVID-19 epidemic crisis. Finally, the suboptimal 25(OH)D levels varied in relation 290 to genetics, age, and disease, so it is important to be cautious when extrapolating the in-291 flection point of 35 ng/mL to other populations or diseases.
Disscusion
Authors write that "notable strengths of the study include relatively sufficient sample size", but according to my knowlaege it is the main aspect of every study to gain relativly importaint results not only conclusions that are not related to the population. Morover authors write that study strengh is that we show prospective standardized, high-quality clinical and laboratory procedures, but according to the medical and clinical chemistry presented method is not the bes tu ensure statistical significante.
Morover authors should try to showe what are others factors that affect presented results and what are other conclusioions.
Author Response
Thank you for the time and effort that they have put into reviewing the previous version of the manuscript. Your suggestions have enabled us to improve our work. We have tried our best to revise our manuscript and provide the point-by-point responses according to your comments and suggestions. Please see the attachment.
Point 1: Introduction
Authors write about the VitD role but not showing any asocieations of its concentraction and human body regulation (how it takeds place, what is it's role etc...but showed in mechnistic forme).
Response 1: Thank you for your suggestions. We added the potential mechanisms of vitamin D deficiency causing overweight/obesity to the “Introduction” part, please check line 56 to 67 on page 2.
Point 2: Introduction
I suggest to showe what is the mains scop of this work and why it is writen - hypothesis and some sugestions????
Response 2: Thank you for your suggestions. Now we clarified the main scope of the work and why it is written, please check line 70 to line 77 on page 2.
Point 3: Methodology
What was a tottal population of infants in presented population and how it was calculated that presented population is enough for the statistical significnte (1,205 six months old infants and 925 of them (76.76%) was falowed)?????
Response 3: According to a previous paper (PMID:31766709), the prevalence of overweight/obesity in infancy was 18.0%. The OR value of VitD for infant obesity was calculated as 1.5. α = 0.05, power=0.80, the lost-to-follow up rate was 20%. We use the PASS software to calculate the number of the study population.
[Proportions--Two Independent Proportions--Test(Inequality)--Tests for Two Proportions(Rations).]
The result showed that we needed recruited 902 participants. Considering the lost-to-follow-up rate(20%), we need 1083 participants. So, our presented population is enough for satistical analysis.
Point 4: Methodology
Why only one of vit D metabolite was messured, what with others???? In some cases it may not showe the hydroxylic activity in nonenzymatic skin hydroxylation and enzymatic reni hydroxylation. Morover I suggest that authors showe what are the associetation beetwen the serum concentraction and proposed tiny layer evaluation of VitD metabolite.
Response 4: Thank you so much for your good questions. At present, the sum up of 25(OH)D2 and 25(OH)D3 or only 25(OH)D3 levels in serum are commonly used in epidemiologic studies to reveal the association between vitamin D status and obesity (or other diseases) in children or adults. Based on previous literature, the metabolites of 25(OH)D2 have been confirmed to have a shorter half-life due to the lower affinity for vitamin D binding protein, implying low bioavailability. In our lab data (unpublished), we also found 25(OH)D3 in serum in infants was a better indicator for other diseases than the sum up of 25(OH)D2 and 25(OH)D3.
Besides, it is 25(OH)D3 which actually be involved in the mechanism of low vitamin D status as a causal factor of obesity. The expression of both vitamin D3 receptors and enzymes responsible for vitamin D3 metabolism in adipocytes depicted a role for the low vitamin D status per se in the development of obesity by modulating adipocyte differentiation and lipid metabolism. (PMID: 28229265)
We have show the associetation beetwen the serum concentraction and proposed tiny layer evaluation of VitD metabolite. We have confirmed the strong correlation between venous and capillary blood concentrations of 25(OH)D in our previous study. Please check the details in this paper (PMID: 32592276).
Point 5: Disscusion
Authors write that "notable strengths of the study include relatively sufficient sample size", but according to my knowlaege it is the main aspect of every study to gain relativly importaint results not only conclusions that are not related to the population. Morover authors write that study strengh is that we show prospective standardized, high-quality clinical and laboratory procedures, but according to the medical and clinical chemistry presented method is not the bes tu ensure statistical significante.
Morover authors should try to showe what are others factors that affect presented results and what are other conclusioions.
Response 5: We agree with your comments. We have revised it in the “Discussion” part of the manuscript and deleted the “Other notable strengths of the study include relatively sufficient sample size, prospective design, and standardized, high-quality clinical and laboratory procedures.”
We considered many factors possibly related to overweight/obesity in our study, including feeding modality, mode of delivery, gestational age, birthweight (kg), passive smoking, outdoor time (h/d), etc. However, we found these factors are not statistically significantly associated with weight-for-length percentile at both monitored times. We agree with you that we didn’t illustrate clearly how these factors may affect the presented results. The probable reason is that the current study was conducted in this population only, which may bias our results. Now we supplement the limitation in the “Discussion” part, please check line 301 to 306 on page 10.
Round 2
Reviewer 1 Report
Ready to be published
Author Response
Thank you again for your suggestions and comments. Your suggestions have enabled us to improve our work.